# Maturity Assessment of Different Table Grape Cultivars Grown at Six Different Altitudes in Lebanon

**DOI:** 10.3390/plants12183237

**Published:** 2023-09-12

**Authors:** Najwane Hamie, Diana Nacouzi, Mariam Choker, Maya Salameh, Linda Darwiche, Walid El Kayal

**Affiliations:** Faculty of Agricultural and Food Sciences (FAFS), American University of Beirut (AUB), P.O. Box 11-0236, Beirut 1107, Lebanon

**Keywords:** table grapes, maturity indices, altitudes, environmental conditions

## Abstract

Table grapes are harvested based on well-known maturity indices that must be monitored after fruit veraison. The aim of this study was to assess these indices across multiple locations and environmental conditions, encompassing different table grape cultivars such as Black Pearl, Crimson Seedless, Superior Seedless, and Red Globe. For this reason, grape sampling was conducted across six distinct locations characterized by varying altitudes above sea level (m asl) and environmental conditions over the ripening season. The main maturity indices, including pH, sugar content, titratable acidity, berry firmness, and other parameters were monitored over the growing season. Moreover, the quantification of total polyphenols, total anthocyanins, and antioxidant activity was determined using spectrophotometric assays at harvesting. The study has examined the effect of the vineyard’s location on grape quality and its interaction with the cultivar and environment. Crimson Seedless maintained a relatively high level of acidity with altitude near harvesting. Black Pearl exhibited a notable decline in both sugar content and berry firmness as elevation increased, whereas Red Globe demonstrated contrasting outcomes. The optimal maturity of Superior Seedless was observed at an elevation of 1000 m asl. Black Pearl and Crimson Seedless exhibited better adaptability to intermediate elevations (650 and 950 m asl), while Red Globe and Superior Seedless showed better adaptability to higher elevations (1000–1150 m asl). Among the studied cultivars, Black Pearl exhibited significantly higher levels of total polyphenols and anthocyanins, while close values were noticed between red and green cultivars.

## 1. Introduction

Grape (*Vitis vinifera* L.) is one of the most diffused fruit crops in the world with almost 78 million metric tons (Mt) of global production [1]. The dominant category with 57% of world production is reserved for winemaking, while nearly 36% are devoted to fresh table grapes and 7% for dried grapes [2]. Grapes are not only known for the good taste of berries and wines but also for the bioactive compounds like polyphenols, which provide various health benefits including antioxidant, anticancer, and cardio-protective effects [3,4,5]. Lebanon is among the oldest countries worldwide growing grape vineyards with a total production of 62,955 metric tons and a total harvested area of 7193 ha [1,6]. Lebanon dedicates 70% of its cultivated area for table grape production and 30% for winemaking [7]. Both local cultivars, including “Beitamouni”, “Tfeifihi”, and “Obeidi”, and imported cultivars, such as Superior Seedless, “Italia”, and Red Globe, are grown in Lebanon and are well adapted to the Mediterranean agroclimatic conditions of this country [7,8,9].

One of the main challenges that table grape growers and researchers encounter is the spatial and temporal heterogeneity of grape ripening in vineyards. Even within the same cluster, grape berries do not ripe uniformly [10,11]. For this reason, the optimal harvesting is considered the most critical point that influences the quality of table grapes in storage and markets [12]. Achieving the optimal harvesting of table grapes requires monitoring maturity indices after berry veraison, the onset of grape ripening [13]. Most researchers agree that pH, total soluble solids (TSS), titratable acidity (TA), the sugar-to-acid ratio (TSS/TA), and aroma are the major maturity indices of table grapes [14]. The sugar content is considered the main indicator of ripeness and the TSS/TA ratio influences berry flavor. For example, the taste of berries could be sour when acidity is higher for the same amount of sugar. Berry firmness reflects the lack of physical and physiological disorders, such as cracking, wilting, sun burning, and insect damages [15]. Furthermore, the assessment of phenolic maturity, which involves measuring total polyphenols and anthocyanins, plays a crucial role in determining the optimal harvesting for wine grapes [16,17]. Quantifying these compounds is also essential for evaluating the quality of colored skin table grapes, as they are abundant in polyphenols and anthocyanins during the harvesting period [13].

All the above-mentioned quality parameters are highly affected by several factors that change from one geographical area to another including cultivar, climate, soil, water status, cultural practices, and maturity stages [16]. For instance, elevation, slope, and orientation will each affect sunlight exposure and wind circulation for a given vineyard location [18]. Most of the previous studies have focused on the location impact on wine grapes rather than table grapes. According to Cureau et al. [19], location is the main factor influencing grape berry and leaf fungal microbiota of wine grape cultivars. Grapes cultivated in different locations can produce different qualities of wines, even if the vineyards are in close areas having similar climates and viticultural practices [18,20]. Moreover, variations in grape composition were found with respect to the total polyphenols and anthocyanins in wine grapes between different locations [20,21,22,23].

The influence of vineyard location and environmental factors on the quality of wine grapes and wine production is widely recognized, but only limited studies have focused on their impact on table grapes. Therefore, the primary aim of this study was to assess the interactive impacts of both vineyard location and cultivar on a range of maturity indices during the ripening season for various table grape cultivars commonly grown in Lebanon and worldwide.

## 2. Material and Method

### 2.1. Plant Materials and Vineyard Locations

The study was conducted on table grapes (*Vitis vinifera* L.). Black Pearl, Crimson Seedless, Superior Seedless, and Red Globe grown in Lebanon (Appendix A). Samples were collected from 6 locations of different elevations above sea level (m asl): El-Qaa (QAA, 650 m asl), Mansourah (MAN, 900 m asl), Zahle (ZAH, 950 m asl), Kfarzabad (KFZ, 1000 m asl), Kfarmeshki (KFA, 1100 m asl), and Baalbeck (BAA, 1150 m asl) (Figure 1). Sampling was conducted on a weekly basis over the maturity season starting from July to October, depending on the cultivar’s season. The geographical location of the vineyards, agricultural practices implemented, and environmental conditions recorded at each location are presented in the Appendix A, respectively.

### 2.2. Physiochemical Parameters

The chemical properties of grape juice were analyzed each sampling day for all cultivars. Six replicates of grape juice were prepared from 20 berries per replicate and then filtered through a filter cloth. The initial pH of the grape juice was measured with a digital pH-meter (HI 2211 pH/ORP, Hanna Instruments, Singapore). Total soluble solids (TSS) were determined with the Pocket refractometer PAL-1 (Atago Ltd., Tokyo, Japan), and the sugar content was expressed as °Brix. The titratable acidity (TA) of grape juice was determined through titration with 0.1 N NaOH up to an endpoint of pH 8.1. Results were expressed as g/L of tartaric acid, a major organic acid found in grape juice.

The physical characteristics of grape berries and clusters were determined. Berry firmness with skin (KgF) was measured using a penetrometer equipped with a 2 mm diameter probe. Cluster weight (g) was measured using a laboratory digital balance (Radwag WTB 2000; ±0.01 g) and a vernier caliper was used to measure cluster length (cm) and width (cm). In order to determine the dry matter (DM), the fresh weight (FW) of 20 berries was recorded before drying them in an oven at 70 °C for 36 h to obtain the final dry weight (DW). DM (%) was then calculated according to the following formula: DM (%) = [(DW/FW) × 100]. 

### 2.3. Phytochemical Analyses

The determination of phytochemicals was carried out on freeze-dried samples. To conduct the subsequent analyses, 0.2 g of each sample was dissolved in 1 mL dimethyl sulfoxide (DMSO).

Total phenolic content of grapes was determined using the BQC Polyphenol Quantification Assay Kit (BioQuoChem, Oviedo, Spain) based on the Folin–Ciocalteu method [24]. This latter oxidizes phenolic compounds to phenolates at alkaline pH resulting in the formation of a blue-colored molybdenum–tungsten complex that can be spectrophotometrically detected at 700 nm. Absorbance measured at this wavelength is directly proportional to the concentration of the phenolic compounds. The standard reference compound for this assay is gallic acid. The assay protocol was applied using the provided reagents and following the manufacturer instructions.

Total anthocyanins content was also determined following the protocol of BQC Anthocyanins Assay Kit (BioQuoChem, Spain) on a 96-well microplate. Anthocyanins pigmentation is pH-dependent and known to be stable in acidic conditions. The red pigment of anthocyanin molecules appears at low pH and becomes colorless at a pH value of 4.5 and above. The BQC Anthocyanins Assay Kit determines the anthocyanin concentration by measuring the absorbance at 510 nm after sample acidification at a pH of 1.0. All assay reagents were ready to use as supplied. After 10 min at room temperature, the absorbance of wells was measured at 510 and 700 nm for anthocyanins detection and turbidity correction, respectively. Results were expressed as cyanidin 3-glucoside equivalents (mg/kg).

### 2.4. FRAP Assay

The antioxidant activity of grapes was determined with FRAP assay (Ferric Reducing Antioxidant Power) that is based on the reduction of Fe^3+^ to Fe^2+^ by antioxidants [25]. The analyses were performed using the BQC Fast FRAP Antioxidant Capacity Assay Kit that applies for microplate spectrophotometric analysis. The reaction mechanism of this assay occurs at an acidic pH where the antioxidants of the sample reduce the colorless ferric (Fe^3+^) to a blue- or violet-colored ferrous ion (Fe^2+^) that shows maximum absorbance at 593 nm. For this analysis, 10 µL of standard or sample were added to 220 μL of reagent A that was supplied with the kit. The mix was incubated sharply for 4 min at room temperature before reading the absorbance of wells at 593 nm. Based on an iron standard curve, the antioxidant capacity of samples was determined and expressed as ferrous equivalents (μM Fe^2+^/kg).

### 2.5. Statistical Analyses

Statistical analysis was undertaken using R studio software (R.4.3.0) to compare maturity indices and evaluated parameters between the different locations by variety and harvest date. One-way and two-way ANOVA were used followed by Tukey’s Honestly Significant Difference (Tukey HSD) test. One-way ANOVA was performed to compare maturity indices between locations at harvest date and between sampling dates for each location for the same variety. Two-way ANOVA was used to compare two factors and explore the interaction between them. The Kruskal Wallis test was used as a non-parametric alternative to ANOVA followed by Dunn’s test. Differences were considered statistically significant for *p* values < 0.05. Principal Component Analysis (PCA) was performed over the harvesting season to observe the clustering behavior of the varieties and correlations between maturity indices.

## 3. Results

### 3.1. Evaluation of Maturity Indices during Maturation

The variations of the key technological maturity indices (pH, TA, and TSS) and berry firmness measurements across different locations and days of the year (Harvest dates) are shown for each cultivar separately (Figure 2, Figure 3, Figure 4 and Figure 5). Generally, as the ripening period progresses, there is an overall increase in pH and sugar content, along with a decrease in acidity. However, distinct variations were observed among locations for a specific cultivar.

Black Pearl grapes were collected from four different locations (QAA, KFZ, KFA, and BAA) with elevations ranging from 650 to 1150 m asl (Figure 2). The average TSS showed an increasing trend throughout maturation but was inversely related to elevation. The highest TSS levels (20.48 °Brix) were found in QAA, and the lowest levels (13.87 °Brix) were found in BAA. The pH fluctuated over the season in all locations. The highest pH (3.56) was recorded in QAA, and the lowest initial pH (2.92) was recorded in KFA. Titratable acidity increased with elevation; the highest TA values ranged between 6.24 g/L in KFZ and 14.33 g/L in BAA. Notably, the TA was higher in KFA and BAA compared to QAA and KFZ. Berry firmness exhibited minor fluctuations across harvest dates in each location; it ranged between 1.68 KgF in KFA and 2.44 KgF in QAA. KFZ tended to have lower fruit firmness values compared to QAA, KFA, and BAA. There was no significant difference for pH, firmness, TA, and TSS in QAA and KFZ between harvest dates.

Crimson Seedless grapes were collected from three locations (ZAH, KFA, and BAA) from 950 m to 1150 m asl (Figure 3). The total soluble content showed a slight decrease with increasing elevation, with average TSSs of 21.3, 20.3, and 19.5 °Brix in ZAH, KFA, and BAA, respectively. TSS was increasing significantly in BAA from 17.2 to 20.8 °Brix between 21 September 2021 and 26 October 2021, while no significant differences were observed in ZAH and KFA. The titratable acidity level was significantly decreasing over maturation in each location but increased with elevation. The lowest and highest TA values were recorded in ZAH (4.64 and 8.52 g/L, respectively). Initial pH did not differ significantly over maturation in each location. It had an average of 3.34 on the whole season. Berry firmness increased significantly in BAA from 1.98 on 21 September 2021 to 2.61 KgF on 26 October 2021, while no significant differences were observed in ZAH and KFA.

The maturation of Superior Seedless was monitored in four locations (QAA, ZAH, KFZ, and KFA) between 650 and 1100 m asl (Figure 4). The sugar content and pH were significantly increasing over maturation in all locations and reached the highest averages in KFZ with 20.2 °Brix on 3 September 2021 and 3.85 on 20 August 2021. The lowest TSS values (9.5 °Brix) were recorded in KFA (1100 m asl) on 27 July 2021. Titratable Acidity levels were decreasing significantly over maturation but had a fluctuating trend among locations. The highest and lowest TA values were recorded in KFA (11.53 g/L and 4.59 g/L, respectively). Berry firmness decreased significantly in ZAH, KFZ, and KFA with maturation and had a fluctuated increasing trend with elevations; the highest values (2.63 KgF) were recorded in KFA whereas the lowest values (1.65 KgF) were recorded in QAA.

The Red Globe ripening seasons were monitored in five different locations from 650 to 1150 m asl (QAA, MAN, ZAH, KFA, and BAA) (Figure 5). The highest TSS content (18.6° Brix°) was recorded in BAA, and the lowest TSS content (9.52 °Brix) was recorded in ZAH. The TSS content was significantly increasing over maturation, while TA was significantly decreasing in each location. However, the highest titratable acidity values were found in ZAH (15.32 g/L) and the lowest valued were found in MAN (4.67 g/L). Consequently, the pH-value increased significantly with the lowering acidity over maturation, but the average pH among locations was fluctuating with elevations. Also, berry firmness decreased significantly with ripeness in all locations. The highest fruit firmness (4.17 KgF) was recorded in ZAH while the lowest fruit firmness (1.72 KgF) was recorded in QAA.

### 3.2. Physiochemical Characteristics of Grapes at Harvest

The quality indices measured at harvest are shown in Table 1. In addition to the parameters presented in Figure 2, Figure 3, Figure 4 and Figure 5 (i.e., pH, TSS, TA and berry firmness); TSS/TA ratio and dry matter were added to evaluate them at harvesting.

The sugar content of Black Pearl was significantly decreasing with elevation among locations, and the highest TSS and TSS/TA ratio were obtained in QAA at 650 m asl with 20.4 °Brix and 27.4, respectively. Dry matter content (20.54%) and berry firmness (2.2 KgF) were more important in QAA as well. Crimson Seedless had a significant increase of acidity with elevation reaching 6.12 g/L in BAA while minor differences were detected for TSS content and berry firmness among locations, averaging 20.9 °Brix and 2.5 KgF, respectively. The highest TSS/TA ratio was obtained in ZAH with 46.1. Dry matter concentrations were significantly different between 950 and 1100 m asl, with an average value of 20.22%. Regarding Superior Seedless, the optimal quality parameters at harvesting were observed at KFZ. The values of pH, TSS, TSS/TA ratio, dry matter, and berry firmness tend to increase significantly with elevation until KFZ, after which they started to decrease in KFA. The maximum TA was 6.70 g/L in ZAH, and then it decreased to 4.59 g/L in KFA where the sugar content averaged 15.1 °Brix. Red Globe had a significant increase of TSS with elevation, but the values of pH and TA were fluctuating with elevations. BAA (1150 m asl) had significantly the highest TSS content, dry matter, and berry firmness with 18.6 °Brix, 20.67%, and 2.1 KgF, respectively. A comparable TSS/TA ratio was obtained at 900 and 1150 m asl with almost 34.7 in MAN and 33.2 in BAA. The overall combined interaction of cultivar and location had a significant influence on all the studied quality parameters of grapes (*p* ≤ 0.001).

### 3.3. Location Effect on Clusters

The effects of the vineyard’s location on cluster weight and size are shown in Table 2. Location and altitude had no significant influence on Black Pearl clusters harvested from different locations, with an average of 520.9 g for weight, 19.2 cm in length, and 13.4 cm in width. Crimson Seedless had significantly longer clusters with increased elevation reaching 20.9 cm in BAA, but no significant differences were detected for cluster width and weight. Superior Seedless had a significant increase in cluster weight and width with elevation, averaging 716.8 g and 13.5 cm in KFA, respectively. Cluster length fluctuated among locations, and the longest Superior Seedless clusters (22.8 cm) were also collected from KFA. The heavier clusters of Red Globe were observed in ZAH (950 m asl) with 909.5 g for weight, 21.8 cm in length, and 13.6 cm in width. In general, the interaction between location and cultivar was more significant for cluster weight and length compared to cluster width.

### 3.4. Principal Component Analysis (PCA)

PCA analysis was conducted at harvest to assess the location effect on the maturity indices of grapes (Figure 6). As expected (Figure 6A), a negative correlation was detected between TA and pH. However, a positive correlation was observed between TSS, berry firmness, and dry matter, as well as between TSS/TA ratio and pH. On the other hand, there was no correlation between cluster length and width (Figure 6B).

### 3.5. Phytochemicals and Antioxidant Activity

Quantification of total polyphenols, total anthocyanins, and FRAP value of the studied cultivars at harvest is reported in Table 3. Black Pearl demonstrated significantly the highest total polyphenols with 946.9 mg GAE/kg. Superior Seedless followed Black Pearl with 672.2 mg GAE/kg but revealed no significant differences with Crimson Seedless and Red Globe, which had an average total polyphenol of 588.9 and 493.7 mg GAE/kg, respectively. Among the seeded cultivars, the polyphenol content in the seeds was significantly higher than in the pulp. Black Pearl and Red Globe demonstrated seed polyphenol levels of 1696.3 mg GAE/kg and 1490.7 mg GAE/kg, respectively, with no significant difference between the two cultivars. Regarding the anthocyanins contents that are expressed as cyanidin 3-glucoside equivalents, Black Pearl had significantly the highest concentration, reaching 1799.3 mg/kg, followed by Red Globe (280.5 mg/kg) and Crimson Seedless (187.1 mg/kg). Superior Seedless had significantly the lowest concentration of anthocyanins with only 6.7 mg/kg. For both Black Pearl and Red Globe, the anthocyanin content in the seeds was significantly lower when compared to the pulps, with values of 64.6 and 60.7 mg/kg, respectively. In parallel with polyphenols and anthocyanins, the antioxidant activity of Black Pearl was the highest among the studied cultivars with 3082.0 μM Fe^2+^/kg, followed by Superior seedless, Crimson Seedless, and Red Globe. The antioxidants of seeds were considerably higher than pulps and were similar in value between Black Pearl and Red Globe with an average of 6781.9 μM Fe^2+^/kg.

## 4. Discussion

Extensive research has examined the various environmental effects on wine grapes and their polyphenolic content [26,27,28,29,30]. However, table grapes have received comparatively less attention, with only a few studies exploring their susceptibility to environmental factors. For instance, Vial et al. [31] found that vineyard location had a greater impact than maturity on the incidence of skin browning of Princess table grapes. However, there have been limited studies investigating the impact of elevation on the physiochemical indices of table grapes. Therefore, this study focused on assessing the ripening period of various table grape cultivars, considering both the vineyard location and environmental factors.

Regarding the environmental impact on grape maturation, cultivars exhibited different ripening patterns in different locations. Based on the environmental data recorded from June to October (Appendix A), it was generally observed that QAA had higher average temperatures (18.4–27.1 °C) compared to ZAH, KFZ, KFA, and BAA. In terms of relative humidity, both QAA and ZAH showed the highest percentages (54.0–64.1% RH), while BAA displayed the lowest average among the locations. Concerning the dew point, QAA also had the highest average (8.9–16.0 °C), whereas BAA had the lowest (4.8–8.1 °C). The remaining locations had relatively similar dew point values. These factors explained the delay in ripening and the extension of season with increasing elevations, as warmer environment is found at lower altitudes. According to Arias, et al. [32], colder temperatures at higher altitudes can extend berry maturation periods. QAA (650 m asl) showed an early harvesting pattern, taking place between August and early September for all cultivars, comparing to other locations where harvesting occurred between late September and October. This outcome was expected, given that QAA had the lowest altitude and the highest average air temperature and dew points among the locations, in addition to a high relative humidity, along with ZAH, during the period from July to October. Moreover, on the same harvest date, lower altitudes exhibited higher sugar contents. For instance, Black Pearl had higher TSS at 650 m asl (18.5 °Brix) compared to 1150 m asl (13.9 °Brix) on 24 August 2021. Similarly, at 950 m asl, Superior Seedless had a higher TSS (13.9 °Brix) than at 1000 m asl (11.8 °Brix) on 29 July 2021. (Figure 4). Black Pearl and Crimson Seedless had a decreasing average TSS and pH as the elevation increased, in parallel with an increase in acidity levels (Figure 2 and Figure 3). On the other hand, the quality parameters of Superior Seedless increased with elevation until they reached optimal values in KFZ at 1000 m asl. Red Globe demonstrated minimal variations across different locations and showed adaptability to various altitudes ranging from 650 to 1150 m above sea level. Superior Seedless exhibited an increase in berry firmness with elevation, while only minor variations were detected for the other cultivars. At 1150 m asl, Black Pearl experienced a notable decrease in berry firmness from 2.4 to 1.83 KgF through maturation, whereas Crimson Seedless had a significant increase from 1.9 to 2.6 KgF. Thus, it can be concluded that the same vineyard altitude has distinct effects on different table grape cultivars. A recent review that assessed the effect of altitude on grape and wine phenolics and volatiles proved that the altitude factor is cultivar-dependent and affect differently the chemical composition (mainly phenolic compositions) of some cultivars that were not included in the present study [30].

Considering the maturation season regardless of vineyard locations, the maturity indices had comparable trends among cultivars over ripening. As expected, the total soluble solids content showed an increase as maturation progressed, while titratable acidity decreased, leading to a rise in pH (Figure 2, Figure 3, Figure 4 and Figure 5). These results confirmed the expected physiochemical changes that happen throughout berry development [33,34]. Berry firmness decreased in Red Globe and Superior Seedless but was slightly fluctuating in Black Pearl and Crimson Seedless. Changes in primary cell wall polysaccharides proved to cause texture changes that result in a decrease of firmness during the ripening of different fruits; however, this feature may differ among cultivars [34,35,36]. For instance, Thompson Seedless reduced firmness from veraison until harvest, but this was not the case of NN107, a newly introduced grape variety that increased in firmness close to harvest [37].

Based on the studied parameters, all cultivars were harvested according to the quality standards and marketable maturity (Table 1). According to the EU Regulation No 543/2011, grapes are harvested at a minimum TSS content of 16 °Brix or 14 °Brix for seedless varieties and a minimum TSS/TA ratio equal to 20 [38]. Typically, the range for titratable acidity (TA) is between 6 and 8 g/L, while the desirable pH level is around 3.1–3.3 for white grapes and 3.3–3.5 for red grapes [39,40]. These factors are important to be considered both to meet the consumer preferences and the export standards for international markets. At harvest, the maturity indices showed different results among cultivars grown at different elevations. Black Pearl had a significant decrease of TSS content and pH, whereas these indices were significantly increasing for Superior Seedless and Red Globe with increased altitudes. In contrast, TA was decreasing with increased TSS, and vice versa. While Crimson Seedless showed only minor variations in TSS across different locations, it displayed a significant increase in acidity levels with higher elevations. In accordance with the PCA results, TSS/TA ratio, dry matter, and berry firmness varied in parallel with the TSS content (Figure 6). Regarding altitude influence on cluster maturation, cluster weight and length changed in parallel with TA and were more affected by location than cluster width (Table 2).

Regarding the phytochemical analyses and antioxidant activity, the overall results indicate that Black Pearl grapes possess the highest levels of total polyphenols and anthocyanins, making them more beneficial in terms of antioxidant properties. The anticipated outcome can be attributed to the distinct characteristics of Black Pearl, which possesses a black-colored skin leading to higher concentrations of anthocyanins compared to the red cultivars, Crimson Seedless and Red Globe. In contrast, Superior Seedless, being a bright yellow/greenish cultivar, exhibited lower overall levels. The total polyphenols of Crimson Seedless (588.9 mg/kg) and Red Globe (493.7 mg/kg) were comparable to the findings of Nicolosi, et al. [41], where values averaged 477.6 and 480.6 mg/kg, respectively. However, this was not the case for Black Pearl and Superior Seedless, as the results from the present study (946.9 and 672.2 mg/kg, respectively) exceeded the values reported in the same study [41] (368.1 and 364.4 mg/kg, respectively). Another study that was conducted on Scarlotta Seedless red cultivar showed a comparable total phenolic content to Red Globe, with an average of 451.5 mg/kg at harvest [42].

In line with the findings, Red Globe exhibited a higher anthocyanin content (280.5 mg/kg) compared to Crimson Seedless (187.1 mg/kg), supporting previous research conducted on red cultivars [43]. Black pearl and Crimson Seedless had higher concentrations of total anthocyanins (1799.3 and 187.1 mg/kg, respectively) compared to another study conducted by de Palma et al. [44], with 1055 and 140 mg/kg, respectively. In terms of antioxidant activity, research has demonstrated that the FRAP value of 30 table grape cultivars with varying colors falls within the range of 1.289 to 11.767 μmol Fe^2+^/g FW (Fresh Weight). The obtained results showed that Black Pearl, which had the highest content of antioxidants, had the highest antioxidant activity (3082.0 μmol Fe^2+^/kg), followed by Superior Seedless, Crimson Seedless, and Red Globe.

It is commonly known that purple/black grapes exhibit higher antioxidant activity compared to red and green cultivars due to their elevated anthocyanin content. However, what is interesting is the presence of comparable FRAP values between red and green cultivars. This phenomenon could be attributed to the higher concentration of flavonols, which are the predominant form of polyphenols in green cultivars, while red cultivars are more concentrated in anthocyanins. These findings align with a previous study conducted by Callaghan, et al. [45]. The concentration of total polyphenols and antioxidants in Black Pearl and Red Globe seeds was expected to be higher than that in the pulp, considering that seeds are known as a rich source of monomeric phenolic compounds and proanthocyanidins, which are the primary polyphenols commonly found in red wine and grape seeds [46,47]. In addition, aroma characterization is an important parameter for table grape quality. According to Wu et al. 2016, fatty and balsamic series are considered favored aromatic series for consumers; therefore, they can be useful indicators for the development of breeding programs and agricultural practices for table grapes [48]. Nonetheless, post-harvest technologies have the potential to extend the harvesting season of table grapes [49]. Researchers recommend repeating the same type of analysis over several years in order to confirm the data and take into account the different environmental factors that may have an impact on fruit quality [50].

## 5. Conclusions

Vineyard altitude is among the environmental factors that affect the maturation of grapes. Since most of the previous studies were conducted on wine grapes and wine phenolics, the objective of this study was to assess the combined effect of cultivar and location on the maturity parameters of the most-grown table grape cultivars worldwide. Overall, it can be concluded that the effect of vineyard location is cultivar-dependent. The quality of the same cultivar can vary significantly when grown in different locations and environmental conditions. Factors such as climate, temperature, humidity, and altitude can all influence the growth and development of table grapes. Black Pearl and Crimson Seedless had a decreasing sugar content, whereas Superior Seedless and Red Globe had an increasing sugar content with higher altitudes. Unlike the other cultivars, Crimson Seedless had an increased berry firmness at 1150 m asl. Cluster weight and length were more influenced by altitudes than cluster width. Therefore, it is essential to consider these environmental factors when evaluating and comparing the quality of grapes grown in different regions, as general conclusions cannot be applied uniformly across different cultivars of table grapes.

## Figures and Tables

**Figure 1 plants-12-03237-f001:**
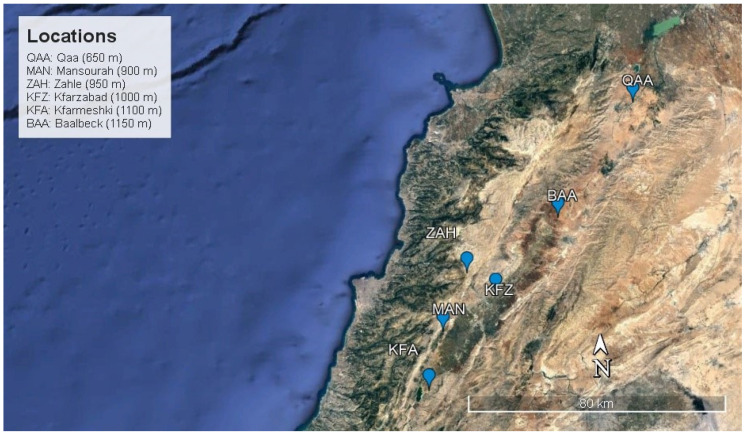
Vineyard locations distributed on the map of Lebanon. QAA (El Qaa, 650 m asl); MAN (Mansourah, 900 m asl); ZAH (Zahle, 950 m asl); KFZ (Kfarzabad, 1000 m asl); KFA (Kfarmeshki, 1100 m asl); BAA (Baalbeck, 1150 m asl).

**Figure 2 plants-12-03237-f002:**
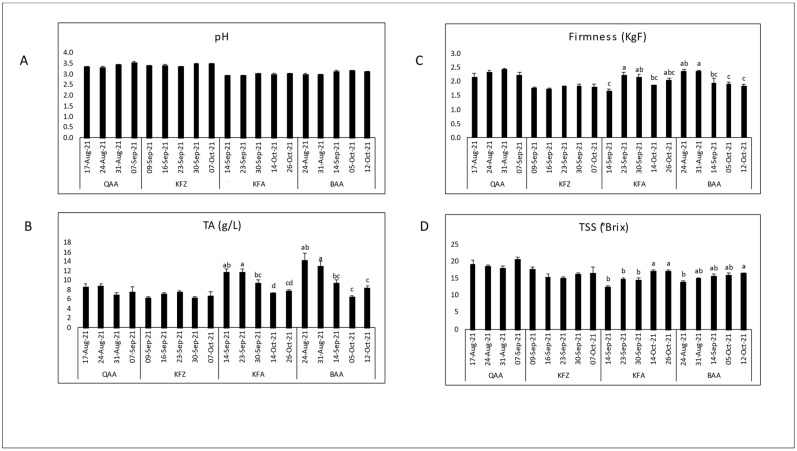
Evaluation of the physiochemical parameters, initial pH (**A**), titratable acidity (**B**), fruit firmness (**C**), and total soluble solids (**D**), through the maturation seasons of Black Pearl table grapes in four locations: QAA (El-Qaa, 650 m asl); KFZ (Kfarzabad, 1000 m asl); KFA (Kfarmeshki, 1100 m asl); and BAA (Baalbeck, 1150 m asl). Means with different letters are significantly different (*p* ≤ 0.05) between harvest dates for each location.

**Figure 3 plants-12-03237-f003:**
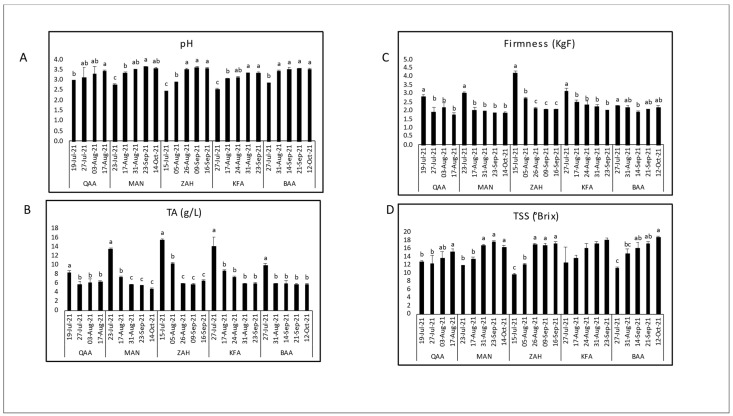
Evaluation of the physiochemical parameters, initial pH (**A**), titratable acidity (**B**), fruit firmness (**C**), and total soluble solids (**D**), through the maturation seasons of Crimson Seedless table grapes in three locations: ZAH (Zahle, 950 m asl); KFA (Kfarmeshki, 1100 m asl); and BAA (Baalbeck, 1150 m asl). Means with different letters are significantly different (*p* ≤ 0.05) between harvest dates for each location.

**Figure 4 plants-12-03237-f004:**
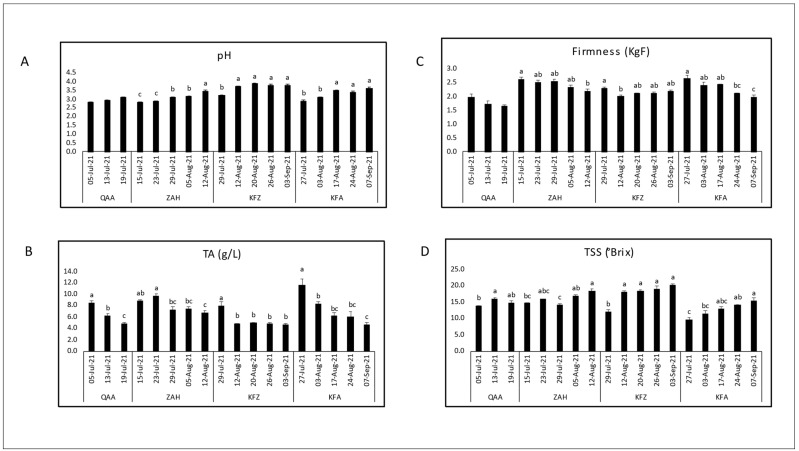
Evaluation of the physiochemical parameters, initial pH (**A**), titratable acidity (**B**), fruit firmness (**C**), and total soluble solids (**D**), through the maturation seasons of Superior Seedless table grapes in four locations: QAA (El-Qaa 650 m asl); ZAH (Zahle, 950 m asl); KFZ (Kfarzabad, 1000 m asl); and KFA (Kfarmeshki, 1100 m asl). Means with different letters are significantly different (*p* ≤ 0.05) between harvest dates for each location.

**Figure 5 plants-12-03237-f005:**
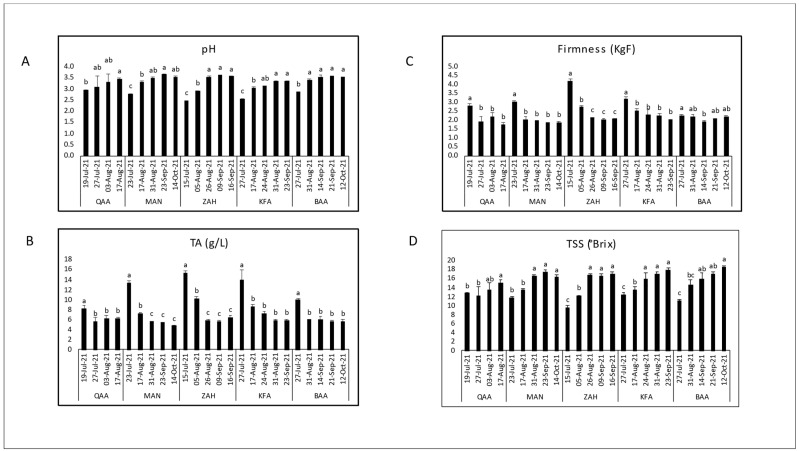
Evaluation of the physiochemical parameters, initial pH (**A**), titratable acidity (**B**), fruit firmness (**C**), and total soluble solids (**D**), through the maturation seasons of Red Globe table grapes in five locations: QAA (El-Qaa, 650 m asl); MAN (Mansourah, 900 m asl); ZAH (Zahle, 950 m asl); KFA (Kfarmeshki, 1100 m asl); and BAA (Baalbeck, 1150 m asl). Means with different letters are significantly different (*p* ≤ 0.05) between harvest dates for each location.

**Figure 6 plants-12-03237-f006:**
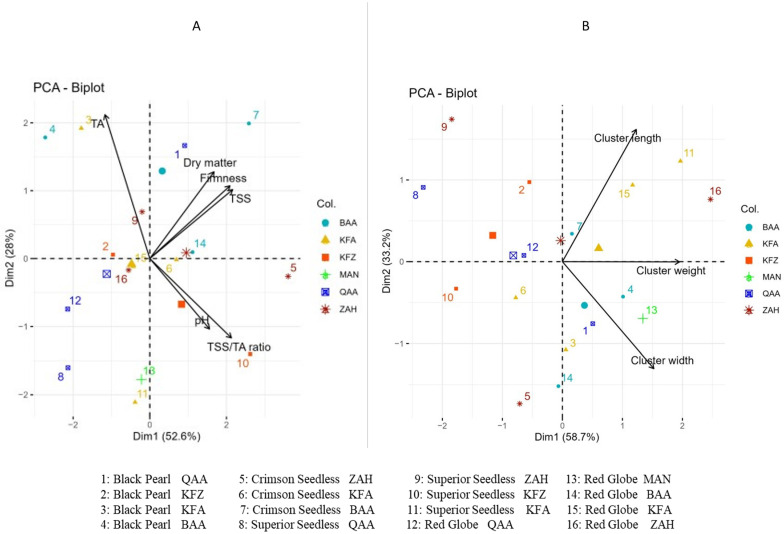
Principle component analysis (PCA) of maturity indices of table grapes (**A**) (TA, dry matter, firmness, TSS, TSS/TA, and pH) and (**B**) (cluster length, cluster width, and cluster weight) harvested from different locations: QAA (El-Qaa, 650 m asl); MAN (Mansourah, 900 m asl); ZAH (Zahle, 950 m asl); KFZ (Kfarzabad, 1000 m asl); KFA (Kfarmeshki, 1100 m asl); and BAA (Baalbeck, 1150 m asl).

**Table 1 plants-12-03237-t001:** Evaluation of the physiochemical parameters of different table grape cultivars at harvest.

Cultivar	Location	Elevation (m asl)	pH	TSS (°Brix)	TA (g/L)	TSS/TA Ratio	Dry Matter (%)	Firmness (KgF)
Black Pearl	QAA	650	3.56 ± 0.03 a	20.4 ± 0.7 a	7.45 ± 1.26	27.4 ± 2.7	20.54 ± 0.89 a	2.2 ± 0.1 a
	KFZ	1000	3.48 ± 0.04 a	18.4 ± 0.6 ab	6.77 ± 0.75	27.2 ± 4.1	17.64 ± 1.72 b	1.8 ± 0.1 b
	KFA	1100	3.05 ± 0.02 c	16.9 ± 0.5 b	7.67 ± 0.20	20.9 ± 1.7	19.01 ± 0.68 ab	2.1 ± 0.1 ab
	BAA	1150	3.11 ± 0.05 bc	16.3 ± 0.4 b	8.36 ± 0.55	19.5 ± 1.5	17.75 ± 0.61 b	1.8 ± 0.1 b
Crimson Seedless	ZAH	950	3.55 ± 0.05 a	21.4 ± 0.3	4.64 ± 0.26 b	46.1 ± 3.4 a	22.27 ± 1.05 a	2.4 ± 0.1
	KFA	1100	3.22 ± 0.02 b	20.6 ± 0.4	5.58 ± 0.24 ab	36.9 ± 2.1 b	14.33 ± 0.31 b	2.4 ± 0.1
	BAA	1150	3.39 ± 0.02 a	20.7 ± 0.7	6.12 ± 0.35 a	33.8 ± 1.5 b	24.06 ± 0.33 a	2.6 ± 0.1
Superior Seedless	QAA	650	3.11 ± 0.03 b	14.5 ± 0.8 c	4.81 ± 0.26 b	30.1 ± 2.9 b	15.85 ± 0.79 b	1.6 ± 0.1 b
	ZAH	950	3.42 ± 0.06 b	18.3 ± 0.8 ab	6.70 ± 0.42 a	27.3 ± 2.1 b	17.90 ± 0.7 ab	2.1 ± 0.1 a
	KFZ	1000	3.78 ± 0.05 a	20.2 ± 0.4 a	4.63 ± 0.16 b	43.6 ± 1.7 a	19.20 ± 0.39 a	2.1 ± 0.1 a
	KFA	1100	3.59 ± 0.06 b	15.1 ± 1.0 bc	4.59 ± 0.38 b	32.9 ± 4.1 ab	15.90 ± 1.13 b	1.9 ± 0.1 b
Red Globe	QAA	650	3.43 ± 0.06 ab	15.1 ± 0.8 c	6.20 ± 0.28 a	24.2 ± 1.9 b	16.19 ± 0.74 b	1.7 ± 0.1 b
	MAN	900	3.52 ± 0.06 ab	16.2 ± 0.5 bc	4.67 ± 0.18 b	34.7 ± 1.4 a	17.35 ± 0.89 b	1.8 ± 0.1 b
	ZAH	950	3.53 ± 0.06 a	16.9 ± 0.6 abc	6.31 ± 0.47 a	26.8 ± 2.1 ab	17.32 ± 0.61 b	2.1 ± 0.1 ab
	KFA	1100	3.32 ± 0.04 b	17.8 ± 0.6 ab	5.82 ± 0.21 a	30.6 ± 1.9 ab	17.75 ± 0.62 b	1.9 ± 0.1 ab
	BAA	1150	3.51 ± 0.04 ab	18.6 ± 0.3 a	5.60 ± 0.39 ab	33.2 ± 1.9 a	20.67 ± 0.74 a	2.1 ± 0.1 a
Cultivar ∗ Location			***	***	***	***	***	***

For each cultivar, means with different letters are significantly different among locations (*p* ≤ 0.05). QAA (El-Qaa); MAN (Mansourah); ZAH (Zahle); KFZ (Kfarzabad); KFA (Kfarmeshki); BAA (Baalbeck); m asl (meters above sea level); TSS (total soluble solids); TA (titratable acidity). *p* values less than 0.001 are summarized with three asterisks.

**Table 2 plants-12-03237-t002:** Influence of vineyard location on cluster size of different table grape cultivars at harvest.

Cultivar	Location	Elevation (m asl)	Cluster Weight (g)	Cluster Length (cm)	Cluster Width (cm)
Black Pearl	QAA	650	609.8 ± 67.5	18.4 ± 1.1	13.7 ± 0.8
	KFZ	1000	472.9 ± 41.7	20.1 ± 0.6	10.9 ± 1.5
	KFA	1100	459.1 ± 61.1	18.2 ± 1.1	14.3 ± 0.7
	BAA	1150	542.1 ± 47.9	20.1 ± 1.1	14.8 ± 0.8
Crimson Seedless	ZAH	950	405.5 ± 52.4	16.4 ± 1.4 b	14.1 ± 1.1
	KFA	1100	374.8 ± 34.1	18.2 ± 0.6 ab	12.7 ± 1.1
	BAA	1150	365.1 ± 45.1	20.9 ± 1.3 a	13.7 ± 0.9
Superior Seedless	QAA	650	199.9 ± 15.4 c	18.7 ± 1.1 ab	9.6 ± 0.2 b
	ZAH	950	292.1 ± 17.8 bc	20.1 ± 0.8 ab	8.9 ± 0.7 b
	KFZ	1000	320.6 ± 37.6 b	17.1 ± 0.9 b	11.1 ± 0.3 ab
	KFA	1100	716.8 ± 24.9 a	22.8 ± 1.1 a	13.5 ± 0.5 a
Red Globe	QAA	650	474.5 ± 49.1 b	18.6 ± 1.1	11.6 ± 1.4
	MAN	900	735.1 ± 71.7 ab	19.1 ± 1.5	14.3 ± 0.8
	ZAH	950	909.5 ± 87.1 a	21.8 ± 0.8	13.6 ± 1.4
	KFA	1100	689.5 ± 32.1 ab	21.3 ± 1.5	12.6 ± 0.5
	BAA	1150	572.2 ± 68.2 b	16.7 ± 1.7	13.7 ± 0.9
Cultivar ∗ Location			***	***	*

For each cultivar, means with different letters are significantly different among locations (*p* ≤ 0.05). QAA (El-Q); MAN (Mansourah); ZAH (Zahle); KFZ (Kfarzabad); KFA (Kfarmeshki); BAA (Baalbeck); m asl (meters above sea level). Means followed by different letters indicate statistical significance at α = 0.05 (*) and 0.001 (***).

**Table 3 plants-12-03237-t003:** Quantification of total polyphenols, total anthocyanins, and FRAP value of the studied cultivars at harvest.

Cultivar	Color	Total Polyphenols (mg GAE/kg)		Total Anthocyanins (mg/kg)		FRAP (μM Fe^2+^/kg)	
		Pulp	Seeds	Pulp	Seeds	Pulp	Seeds
Black Pearl	Black/purple	946.9 ± 17.7 a	1696.3 ± 70.7 A	1799.3 ± 15.4 a	64.6 ± 3.1 B	3082.0	6799.6
Crimson Seedless	Red	588.9 ± 77.6 b	-	187.1 ± 21.3 c	-	1878.9	-
Superior Seedless	Bright yellow/greenish	672.2 ± 98.3 b	-	6.7 ± 7.1 d	-	2053.6	-
Red Globe	Red	493.7 ± 18.1 b	1490.7 ± 141.4 A	280.5 ± 4.7 b	60.7 ± 1.5 B	1613.5	6764.2

Significant differences among cultivars are indicated by lowercase letters, while significant differences between pulp and seeds within each cultivar are indicated by uppercase letters (*p* ≤ 0.05).

## Data Availability

Not applicable.

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
