# Peer review of "Maturity Assessment of Different Table Grape Cultivars Grown at Six Different Altitudes in Lebanon"

_plants, 2023, doi:10.3390/plants12183237_

Round 1

Reviewer 1 Report

The MS“Maturity assessment of different table grape cultivars grown in different locations” is interesting, it provides the fruit quality parameters under different growth location.

1.     Are the cultivation management and soil quality consistent in different regions.

2.     The same number of days for different regions of the same variety should be taken, in order to compare their difference in the same sampling date.

3.     SD is missing in figure 1 and 2, and table 2.

4.     Table1 has some repeat with the figures data.

Author Response

Thank you for your positive and valuable evaluation. We are pleased that you found the study's focus on table grape quality parameters across various growth locations to be interesting. Please find herewith our response to your comments and suggestions.

Are the cultivation management and soil quality consistent in different regions.

The cultivation management was quite similar among the different locations. The adapted training system for the vines was Pergola, except for the QAA location. Clay soil was the dominant soil type, except for KFA, where silt soil dominated. Further details, including agricultural practices, can be found in supplementary Table S2. 

The same number of days for different regions of the same variety should be taken, in order to compare their difference in the same sampling date.

While having uniform sampling dates would have been ideal, geographical variations made it impractical to collect samples simultaneously from all locations. In practical terms, harvesting grapes from different regions on the same day is a time-consuming task, especially given our objective of assessing their quality immediately without the use of cold storage. Nevertheless, whenever we were able to collect samples concurrently, we compared the results in the manuscript (Line 348-349: Moreover, at the same 348 DOY, lower altitudes exhibited higher sugar contents. For instance, Black Pearl had higher 349 TSS at 650 m asl (18.5 °Brix) compared to 1150 m asl (13.9 °Brix) on DOY 236 (24 August 2021). Similarly, at 335 950 m asl, Superior Seedless had a higher TSS (13.9 °Brix) than at 1000 m asl (11.8 °Brix) 350 on DOY 210 (29 July 2021) (Fig. 2).

SD is missing in figure 1 and 2, and table 2.

The standard deviation (SD) was incorporated into all figures; however, in cases where values were small or negligible, they might not have been visibly displayed on the bars. While we did not include the SD into the captions, we will ensure its addition to all figures. Additionally, we will enhance the resolution of the figures to ensure that the SD can be clearly presented.

Table1 has some repeat with the figures data.

The figures showed the measurements of pH, TSS, TA and firmness throughout the entire maturation season, whereas the table provided supplementary data specifically at the time of harvesting. Given the variation in Day of Year (DOY) across different cultivars and locations, the table served to summarize and emphasize the optimal maturity stage for each cultivar and to compare easily among altitudes at the same maturity level (harvest time).

Reviewer 2 Report

Grape quality is effected by many factors such as variety, climate, maturity. In this paper, the researchers analyzed the maturity indices of Black Pearl, Crimson Seedless, Superior Seedless, and Red Globe at different altitudes, which to a certain extent provides theoretical guidance for the regulation of grape quality and this is the first time to focus the effect of altitude on table grape quality in this region. However, there has been a lot of researches focusing the influence of altitudes on grape quality, including basic physical and chemical indicators, phenolic substance, aroma and so on. This paper only analyzed the simple physiology indicators of grapes. What about other maturity indices that possibly contributed to the monitor, such as grape taste, grape aroma and so on? It will be interesting to see the difference in the taste and aroma of grapes from different altitudes.

The climate between different altitudes and locations were different. The influences of altitudes and related environment on grape maturity are not further explored. How were the differences correlated with the grape quality? The experimental design was relatively simple and the data analysis was not in-depth enough. What is the mechanisms under the data?

Line 79, Provide the longitude and latitude for Lebanon.

Line 83, which year was the experiment performed? How many grapes were collected? Is there any replication? What’s the standard for sample collection? Provide more details for the sampling.

Line 103, How many clusters were used for cluster weight? I assume 20 berries were used for firmness.

Line 259, in Table2, the part date of Black Pearl, Crimson Seedless and Red Globe didn’t show the significance analysis results (different letters) and we can’t get to know whether the grape indicators in Table2 were different at different altitudes.

Line 274, based on the results of PCA analysis, the quality of this analysis method is not good and the cumulative variance contribution rate of PCA1 and PCA2 is only 0.591, which didn’t meet the mathematical statistical requirements (above 0.80)

Line 178, there is a problem of error syntax, for example, the sentence of ‘while no significant differences observed in ZAH and KFA” absences predicate, and should be corrected as “while no significant differences were observed in ZAH and KFA’.

Line 228, there is a problem of unclear expression, for example, ‘the sugar content of Black Pearl was significantly decreasing with elevation among locations’, it may be ‘the sugar content of Black Pearl was significantly decreasing with elevation increasing.

Author Response

Thank you for your valuable input. We appreciate your thoughtful insights on our paper. We acknowledge your point regarding the various factors influencing grape quality and the broader range of indicators that can be considered. Indeed, there have been numerous previous studies assessing the effect of altitude on the quality of grapes. However, these studies were mostly conducted on wine grapes. Very few studies have examined the impact of altitude on table grapes, especially focusing on the specific cultivars included in this study.

Additionally, your suggestion to incorporate aroma and taste analyses would have been of great interest. Nevertheless, the study predominantly centered around the technological maturity indices commonly utilized by farmers and grape specialists. Most researchers consider the total soluble solids, titratable acidity and TSS/TA ratio, as the primary indicators to measure grape maturity. These indicators are crucial in determining the optimal harvest time (Piazzolla et al., 2017; http://dx.doi.org/10.4081/jae.2017.639). This concept is known as technological maturity and is widely applied as a standard practice to determine harvesting (Nogales-Bueno et al., 2014; https://doi.org/10.1016/j.foodchem.2013.12.030).

Even internationally recognized regulations and licenses for patented cultivars primarily establish standards, particularly for the TSS content or TSS/TA ratio, to guide the harvesting process. For example, according to the EU Regulation No 543/2011, grapes are harvested at a minimum TSS content of 16 °Brix or 14 °Brix for seedless varieties and a minimum TSS/TA ratio equal to 20. Moreover, the phenolic maturity, primarily represented by the total polyphenol content, was also assessed in the study to better evaluate the quality of the colored varieties.

The climate between different altitudes and locations were different. The influences of altitudes and related environment on grape maturity are not further explored. How were the differences correlated with the grape quality? The experimental design was relatively simple and the data analysis was not in-depth enough. What is the mechanisms under the data?

Given that we conducted the experiment under field conditions across various locations, addressing them within different environmental conditions was quite challenging. Our primary focus was to analyze the impact of altitude and how identical maturity indices exhibited diverse changes with fluctuations in altitude. In the second paragraph of the discussion section, we examined the variations in climatic conditions among the different locations. We focused mainly on temperature, humidity and the dew points as they are the main parameters varying among locations (Supplementary Table S3).

Line 79, Provide the longitude and latitude for Lebanon.

Lebanon is located at approximately 33.8547° N latitude and 35.8623° E longitude. We will include this geographical information in the revised manuscript for clarity.

Line 83, which year was the experiment performed? How many grapes were collected? Is there any replication? What’s the standard for sample collection? Provide more details for the sampling.

Thank you for this comment. We will clarify all the sampling information in the M&M section. The experiment was conducted in 2021. Sampling was carried out on a weekly basis from July to October. On each sampling day, 10 clusters per cultivar were collected from both sides of the specific grapevines. For each cultivar, pH, TSS, and TA were analyzed with 6 replicates on grape juice. The juice was prepared by blending and filtering 20 berries per replicate, randomly selected from the clusters.

Line 103, How many clusters were used for cluster weight? I assume 20 berries were used for firmness.

A total of 10 clusters were used for weight measurements. Regarding firmness, the mean of 10 randomly selected berries per cultivar was determined. We will also add this information to M&M.

Line 259, in Table2, the part date of Black Pearl, Crimson Seedless and Red Globe didn’t show the significance analysis results (different letters) and we can’t get to know whether the grape indicators in Table2 were different at different altitudes.

All the data presented in Table 2 were statistically analyzed; however, in cases where no significant differences were observed, corresponding letters were omitted. To ensure clarity for readers, we will include the appropriate letters in all instances.

Line 274, based on the results of PCA analysis, the quality of this analysis method is not good and the cumulative variance contribution rate of PCA1 and PCA2 is only 0.591, which didn’t meet the mathematical statistical requirements (above 0.80).

We appreciate your attention to detail and your guidance to enhance the PCA analysis. The revised analysis has yielded improved results, addressing the concern you raised regarding the cumulative variance contribution rate.

  • Comments on the Quality of English Language

 Line 178, there is a problem of error syntax, for example, the sentence of ‘while no significant differences observed in ZAH and KFA” absences predicate, and should be corrected as “while no significant differences were observed in ZAH and KFA’.

Thank you for pointing out the error in the syntax on line 178. The sentence has been revised as per your recommendation, ensuring proper syntax and clarity.

Line 228, there is a problem of unclear expression, for example, ‘the sugar content of Black Pearl was significantly decreasing with elevation among locations’, it may be ‘the sugar content of Black Pearl was significantly decreasing with elevation increasing.

As per your suggestion, this sentence has been rephrased in accordance with your recommendation. Thank you.

Reviewer 3 Report

Dear authors and Editors, 

the paper entitled "Maturity assessment of different table grape cultivars grown in different locations" is interesting, table grape production/research was neglected, and now is gaining importance and interest in many countries (Italy, Brazil, Japan, China...), so the topic is very relevant and appreciated.

Vineyard location, elevation and environmental conditions are important and complex to discuss and challenging to make firm conclusions based on one season, however, this shortage is covered with multiple locations, few cultivars, and represent valuable preliminary results. More unifomity of studied sites conditions is desirable, but this research is conducted in a field conditions which is always a challenge and may be understandable.  In supplementary material, there are also some differences in maintaining vineyards (e.g. sporadic application of gibberelic acid (QAA, ZAH), which according to some - may influence some of results - so this needs to be  explained in discussion, if relevant). In Discussion section, main points need to be highlighted and result-reporting shortened.

a slight technical adjustment of the text, tables, spelling need to be checked.

Please find a few comments in the attached pdf.

Recommendation: Minor revisions.

Wish you all the very best, 

Kind regards, 

Reviewer

English seems normal and understandable, however, I'm not a native or professional speaker. Possibly some sentences might be shortened probably by using more professional english phrases and terms.

Author Response

Thank you for your valuable feedback and positive assessment of our paper. We appreciate your recognition of the increasing importance of research in table grape production across various countries. We acknowledge the complexities tied to vineyard location and environmental conditions, as discussed within a single season, and we are pleased that our study's approach, encompassing multiple locations and cultivars, offers preliminary insights.

Regarding the introduction, the main challenge faced by table grape growers that aligns with our objective is mentioned in the second paragraph of the introduction, along with the corresponding citations. The challenge regards the heterogeneity in the field which makes optimal harvesting decisions more complicated and affects the market. In response to your comment, we added another challenging factor, which is related to the physiology of the grapes, along with the corresponding citations. It is important to note that there are very few studies reporting the situation of grape production in Lebanon.

We agree that some results may appear repetitive, but this is due to the parameters displaying similar overall trends. We tried to highlight even minor variations among locations and cultivars. For instance, despite slight changes in TSS in Superior Seedless, it is important to mention them, as TSS is the primary maturity indicator for grapes.

We appreciate the observation regarding your point about differences in vineyard maintenance, such as the application of gibberellic acid.  According to the reference you provided, GA3 application may influence the length of the clusters. It is an important point to consider; however, it was not applicable to our case. As indicated in Table2, QAA and ZAH, the only two locations with GA3 application, did not show significant differences of cluster size compared to other locations. In response to your comment on Black Pearl grapes, the minor variations among different locations and environments actually point to its ability to adapt to diverse climates, largely attributed to its thick, dark skin.

The PCA graph will be completely changed because we have to repeat the analysis. We will try to present it in a clearer way. For the annotation of the PCA, we followed your suggestion of writing each single cultivar in one separate column.

The additional information regarding the hybridization of cultivars is undoubtedly important. However, it is more closely linked to fundamental aspects of genetics and plant breeding, which do not constitute the primary focus of our study.

In response to your feedback, we will highlight key discussion points , present the results more straightforward and concise our result reporting. We will also make the necessary technical adjustments, review spelling, and address all the comments that you have provided in the attached PDF.

English seems normal and understandable, however, I'm not a native or professional speaker. Possibly some sentences might be shortened probably by using more professional english phrases and terms.

Well noted with thanks. We will check the clarity of some complex and long phrases and rephrase it in a more professional and scientific way.

Reviewer 4 Report

The manuscript describes grape maturity in table grape cultivars (cvs. Black Pearl, Crimson Seedless, Superior Seedless, and Red Globe) at six different locations in Lebanon. The experiment was conducted in a single growing season. Conventional indicators of grape maturity (TSS, TA, pH and firmness of berries) were used to monitor grape maturity at different periods from veraison to technological maturity. At the time of technological maturity, the dimensions of the clusters and the physiochemical composition of the grapes were analyzed.

The objective of this study was to determine the influence of location and variety on grape maturity indices. The methodology is confusing and insufficient to test the research objectives. The authors did not present clear evidence of the influence of variety and location on the variables studied. Conventional indicators of grape maturity from different sites and with different DOY are inconsistent. Not all four varieties were studied at the six sites examined (Supplementary Table S1). The exception is the Kfarmeshki site, where all four varieties were analyzed consistently. The applied statistical analysis is not suitable to evaluate the influence of both factors and their interaction. In Table 2, the authors present the results of the interaction, assuming that the data were analyzed by a two-way analysis ANOVA, but without clearly showing the influence of variety and location. In the M&M section, the authors did not mention two-way analysis ANOVA for the data analysis. The results are presented very clumsily. In Figs. 2, 3, 4, and 5, the presentation of results for different variables (pH, TSS, TA, Firmness) expressed in different scales on the same graph is confusing and not easy to follow. The day of the year (DOY) is not consistent for the sites and it is not easy to compare the indices between sites. Statistical significance for each variable is not properly explained in capture. Conclusions were drawn based on the results presented with an inadequate statistical model.

The evaluation of grape maturity to determine the optimal harvest date is an important research topic, especially for table varieties for which experimental data are lacking. This study needs to be reconstructed and reanalyzed with an acceptable statistical model.

In my opinion, the manuscript is not suitable for publication in this form.

Minor comments:

The title is too general and does not tell the reader what the research is about. The authors need to provide some details about location; for example Maturity assessment of different table grape cultivars grown at six different altitudes in Lebanon

Abstract: “Main results confirmed that the effect of vineyard location on the maturity indices is cultivar dependent” - this sentence is too general and speculative. The authors must concretize the results and support them with statistical analyses.

Fig. 1: The source of the map should be indicated according to copyright rules

Material and Methods: Sampling of berries is not clear. The authors stated “six replicates of grape juice were prepared from 20 berries per replicate”, later in M&M the authors stated that cluster weight (g) and cluster length (cm) were also measured. Did you use the same clusters to take 20 berries to measure the maturity indices? How many clusters did you take from each variety at each sampling? In the Results section, it is obvious that cluster dimensions was measured at full maturity, but this information is missing in the M&M section.

Line 143: Provide the full name of the statistical software used for the analysis. The appropriate reference for the statistical software should be included in the reference list 

Author Response

Thank you for your comprehensive evaluation of our manuscript. We greatly appreciate your insights into the methodology, presentation of results, and statistical analysis. Your feedback has highlighted critical areas for improvement in our study. We have taken careful steps to reevaluate and reconstruct the approach to presenting our results and conducting the statistical analysis, aiming to provide a clearer understanding of how variety and location influence grape maturity.

Regarding the methodology, we acknowledge the potential confusion due to our study encompassing four different cultivars, each with varying growing seasons and characteristics across six locations at different sampling times. In practical terms, especially given that we are conducting the experiment under field conditions, it is quite challenging to ensure scientifically robust results due to the variations in multiple factors. Assessing grape quality has always been challenging due to the diverse factors that influence defining optimal maturity, which also differs among cultivars. We believe these results will benefit grape growers and scientists not only in Lebanon but also in other countries. This seemingly fundamental information fills a gap since no previous study has specifically concentrated on the growing season of these commonly cultivated table grape varieties.

Concerning the inconsistency of results across DOYs and locations, we reconstructed the graphs to present the findings with greater consistency. Our intention was to gather samples from commercial vineyards featuring all studied cultivars; however, regrettably, this was unattained due to the unavailability of certain cultivars at the six examined sites (excluding the Kfarmeshki site). This factor was beyond our control.

The significance of the interaction between cultivar and location was statistically analyzed and is shown in Table 1 (last line). Overall, the statistical analysis was repeated to better illustrate the influence of both factors along with their interaction. The specific type of ANOVA used and other statistical analyses will be clearly described in the M&M section.

Minor comments:

The title is too general and does not tell the reader what the research is about. The authors need to provide some details about location; for example Maturity assessment of different table grape cultivars grown at six different altitudes in Lebanon

Thank you for your suggestion regarding the title. We appreciate your feedback and agree that providing more specific information about the research focus is important. We considered your suggestion and the title was modified as per your suggestion

Abstract: “Main results confirmed that the effect of vineyard location on the maturity indices is cultivar dependent” - this sentence is too general and speculative. The authors must concretize the results and support them with statistical analyses.

We understand that it appears as a general statement; nevertheless, we have employed it as an introductory phrase before presenting the main findings in detail. In the abstract, we will provide the specific values obtained and support them with the corresponding statistical analyses.

Fig. 1: The source of the map should be indicated according to copyright rules

Thanks for pointing out this missing information. We appreciate your attention to detail. The map was created using Google Earth Pro version 7.3.6, and we will ensure to include the proper source attribution in compliance with copyright regulations.

Material and Methods: Sampling of berries is not clear. The authors stated “six replicates of grape juice were prepared from 20 berries per replicate”, later in M&M the authors stated that cluster weight (g) and cluster length (cm) were also measured. Did you use the same clusters to take 20 berries to measure the maturity indices? How many clusters did you take from each variety at each sampling? In the Results section, it is obvious that cluster dimensions was measured at full maturity, but this information is missing in the M&M section.

In the revised version, we included a detailed account of the berry sampling process in M&M. On each sampling day, we collected 10 clusters from each variety and location. This juice was obtained by blending and filtering 20 berries per replicate, which were selected randomly from the clusters. To measure cluster sizes, we utilized all the 10 clusters as replicates. Cluster dimensions were tracked throughout the entire season; however, for the purpose of presenting the main data, only the sizes of clusters at harvest were reported. This information will be incorporated into the M&M section.

Line 143: Provide the full name of the statistical software used for the analysis. The appropriate reference for the statistical software should be included in the reference list.

The full name of the software was mentioned in the statistical analyses section. We utilized R 4.2.1 software, a widely recognized statistical tool. Our approach aligns with common convention, where the software name is mentioned without a reference citation, as it primarily serves as a means to access the software rather than for bibliographic citation purposes.

Round 2

Reviewer 2 Report

The manuscript was improved. But I still suggested to add more data to support the conclusion. I understand the basic physiochemical parameters are important. The industry employs TSS, TA, Brix a lot. But it is a research paper published on the research journal, not on industry newspaper. You have to provide sufficient information for all the readers. Aroma is very important parameter for table grape. 

Different color can be applied to the changed part of the manuscript. It's hard to check now. 

The authors have to revise the language of all the manuscript, not only where I mentioned. 

Author Response

We extend our great appreciation to the reviewer for the exceptionally valuable suggestions. Notably, we have introduced a new paragraph (lines 429-436) to delve into the important role of aroma in the ripening of table grapes.

Reviewer 4 Report

The authors have made significant improvements to the manuscript. The revised title now accurately reflects the experimental data presented in the paper.

Furthermore, the manuscript has undergone substantial refinement through methodological corrections. The authors have addressed issues related to the sampling process and improved the statistical analysis. They have also rectified inconsistencies in the results across different days of the year (DOYs) and various locations. These revisions have effectively eliminated doubts that may have arisen from the initial version of the manuscript. I recommend accepting this manuscript in its current form after addressing a minor technical mistake.

I would like to offer a minor suggestion concerning the abstract. The sentence, "Main results confirmed that the effect of vineyard location on the maturity indices is cultivar dependent," might be somewhat misleading. Previous research has identified multiple factors influencing grape maturity indices and grape quality in general (please refer to the review article by Poni et al. 2018: "Grapevine quality: A multiple-choice in Sci Hort"). Therefore, I recommend revising the sentence for greater clarity.

Author Response

We express our gratitude to the reviewer for dedicating their time and effort. As per the reviewer's recommendation, we have made adjustments to the sentence, "Main results confirmed that the effect of vineyard location on the maturity indices is cultivar dependent." The revised sentence in the abstract (lines 20-21) now reads, "This study has investigated the impact of vineyard location on grape quality, considering its interplay with the cultivar and environmental factors."